# Does the South-to-North Water Diversion Project promote the growth of enterprises above designated size in the water-receiving areas?—Evidence from 31 provincial-level administrative regions in China

**Ting Wang**[1], **Jianyu Chi**[2]*

**1** State Key Laboratory of Media Convergence and Communication, Communication University of China, Beijing, China, **2** School of Economics and Management, Communication University of China, Beijing, China

* chijianyu@163.com

**Data Availability Statement:** All relevant data are within the manuscript and its Supporting information files.

## Abstract

The South-to-North Water Diversion Project (SNWDP) is believed to drive the next phase of sustainable productivity growth, meeting growing water demand, so as to address increasing environmental sustainability challenges. The Middle Route of SNWDP is regarded as an extremely large long-distance inter-basin water diversion project, which has benefited Henan, Hebei, Tianjin and Beijing since 2014 with great sustainable changes to the cities, groundwater, ecological environment, industrial structure and social development of the beneficiary areas. Taking the number of industrial enterprises above designated size (IEDS) in the water-receiving areas as the research object, this paper takes the year of policy implementation 2014 as the basic time point, evaluating the change of the number of IEDS in the beneficiary areas of the Middle Route of SNWDP through difference-in-difference model. The results show that: (1) The Middle Route of SNWDP promotes the additional growth of the number of IEDS in the beneficiary areas. (2) When the Middle Route of SNWDP promotes the growth of the number of IEDS in beneficiary areas, there is no regional difference for regions with different development levels. (3) The reasons why the Middle Route of SNWDP contributes to the additional growth of the number of IEDS are composed of promoting mixed ownership reform of enterprises in beneficiary areas, increasing water supply and increasing population. However, the Middle Route of SNWDP has not had a significant impact on the traditional total factor productivity or the components of production factors, technology and capital. From the final outcome, the South-to-North Water Diversion Project has played a facilitating role in the sustainable development of large-scale enterprises in the water-receiving areas.

## Introduction

Freshwater can provide guarantee and impetus for sustainable social and economic development. Water-related nexus and environmental sustainability is difficult to achieve. However,

**Funding:** The author(s) received no specific funding for this work.

**Competing interests:** The authors have declared that no competing interests exist.

population increase, urbanization, climate change, and inadequate governance pose significant threats to freshwater ecosystems worldwide in the twenty-first century [1, 2]. China has problems with both water resources and population distribution, both of which exacerbate water inequality. The southern part of China is relatively rich in water resources, while the Huang-Huai-Hai basin in the north is quite arid [3]. More than one-third of China's population lives in the densely populated northern Huang-Huai-Hai basin, which includes the megacities of Beijing and Tianjin [4]. In this area, the available water supply is less than 400 cubic metres [5]. The uneven distribution of water resources makes the development of enterprises in the North face greater challenges than those in the South [6, 7]. In China, as the world's largest developing country, one of the key sustainability strategies to deal with water scarcity and environmental issues is to provide better infrastructure technology to achieve inter- and intra-basin water resources transfers, especially in arid regions, where such initiatives are more necessary [8, 9]. Therefore, China started the South-to-North Water Diversion Project.

Three routes constitute up the SNWDP. The Eastern Route was a reconstruction and upgrading of the Grand Canal developed during the Sui and Tang Dynasties, which consisted of diverted rivers and inland lakes [10, 11]. In the past, this canal was used by traders along the Yangtze River in China to move products from the Yangtze River basin to the Yellow River basin; The SNWDP's east route's water transfer volume surpassed 5.288 billion cubic meters in 2022. This number is predicted to increase to 14.8 $km^3$/year by 2030. This waterway may soon play the same role it had in the past, simultaneously transporting scarce water resources to the arid northern regions [12, 13]. In contrast, the focus of this discussion in this article—the Central Route project completed in 2014—is a new construction project. It starts from the Danjiangkou Reservoir on the Han River, a tributary of the Yangtze River, and passes through Henan and Hebei provinces, ultimately reaching its destination in Beijing and Tianjin. After blending with water from the recipient areas in a certain proportion, it enters every open tap in the recipient area through water stations [14, 15]. This change is visually reflected in the changes in the water supply to the northern regions after the South-to-North Water Diversion Project began operation.

Due to the significant importance of water resources for industrial production, large-scale infrastructure projects like the South-to-North Water Diversion, are considered capable of driving the next phase of productivity growth. By directly transferring water, it aims to meet the growing water demands of businesses, enabling them to expand their production scale and addressing the increasingly challenging environmental issues [9]. Such projects can enhance the availability of resources such as energy, irrigation and water supply, thereby improving people's living standards [16]. Two of the three routes of the SNWDP—the eastern and central routes—have been operationalized after 60 years of planning. The project has provided the northern region with 53.1 billion cubic meters of water as of May 2022 [17]. Compared to other methods of water resource redistribution, such as rainwater harvesting and water resource reuse, the South-to-North Water Diversion is more suitable for large-scale water resource redistribution, meeting the needs of China's vast arid regions [18].

However, while this direct retrieval of water resources can quickly alter the water supply in the recipient areas, addressing the sustainability challenges arising from water scarcity is rapid and direct. As a significant national project, the South-to-North Water Diversion Project comes with a high cost. Over the years, there has been significant taxpayer interest in whether the project can achieve its goal of driving economic development in practice and the mechanisms through which the project affects the economy in the recipient areas [10]. To address this contemporary need, scholars like Hu and others have discussed the cost changes resulting from the impact of the South-to-North Water Diversion project from the perspective of engineering safety, particularly the cost increase caused by ground subsidence resulting from the

project [19]. On the other hand, some researchers have undertaken a comprehensive estimation of the project's costs using engineering cost estimation methods, such as estimating the environmental costs incurred by the South-to-North Water Diversion project [20]. They have also indirectly assessed the costs of the project by evaluating its impact on different types of ecosystems [21, 22]. These cost-based studies generally agree on the viewpoint that the South-to-North Water Diversion Project successfully shifted the aquatic ecosystems from the water supply area to the recipient area. However, there is a significant disparity in the estimation of investment in the construction project, leading to differing evaluations of the project. In addition to these discrepancies in project costs, there has also been debate over the rationality of the project's benefits. The academic community's divergent opinions mainly focus on the change in the value of water resources brought about by the improvement in water use efficiency. A report from the World Bank suggests that the project has economic benefits, while a report from the World Wildlife Fund argues that the project lacks economic benefits [18].

Beyond the debates about the balance of benefits and whether the benefits in the recipient areas are positive, there has been relatively little academic analysis regarding the mechanisms of economic promotion in the recipient areas of the South-to-North Water Diversion Project. This study focuses on the post-completion changes in capital, technology, and labor due to the project, exploring the different roles of general factors in economic growth influenced by water resources in the recipient regions. Moreover, China, especially in the northern regions, has a relatively high proportion of state-owned enterprises compared to other countries worldwide. In recent years, China has actively been implementing reforms in corporate ownership to address the issue of state-owned enterprises crowding out private enterprises in the market. This is a significant development in the field of corporate economics. However, there has been limited scholarly analysis from this perspective on the impact of large-scale water infrastructure projects.

Double-difference models offer distinct advantages when studying policy effects with clear implementation time points. They can help eliminate the effects of individual fixed factors and time-fixed factors, resulting in more credible causal inferences compared to direct comparisons of before and after treatment rates. To explore this issue from the perspective of applied economics, this research is anchored in company-level data for large-scale industrial enterprises. It employs double-difference models to observe the practical contributions of the South-to-North Water Diversion Project and investigate the mechanisms behind these contributions. Additionally, heterogeneous analyses are conducted to analyze the responses of different enterprises to the project after its completion, demonstrating the favorable impact of the project on corporate ownership structure reform. The aim is to reveal the positive impact of infrastructure development on economic growth and provide valuable insights for future policy formulation in a similar context.

## Literature review

### Importance of water resources to IEDS

In China's practice, IEDS have great advantages in economic growth, industrial development and technological innovation out of their unique characteristics of high labor productivity, high market share of leading products, strong independent R & D capability and advanced technical equipment [23, 24]. Industrial enterprises refer to the material production departments or enterprises engaged in the exploitation of natural resources, processing and reprocessing of extracted products and agricultural products [25, 26]. Since 2011, the starting point standard of IEDS has been raised from the original annual main business income of 5 million yuan to the annual main business income of 20 million yuan [25]. Based on prior research, the

advantages of industrial businesses larger than the designated size in promoting economic growth are becoming increasingly clear, and the pulling effect is growing. They also play a significant role in social and economic development [23]. It is clear that one of the most crucial steps to further support China's economic development is to accelerate the cultivation and development of industrial companies over the designated size.

Marshall argued that the initial agglomeration of businesses in a particular region is driven by natural endowments and policy orientation, meaning that regions with better natural resources and policy support are more likely to nurture competitive enterprises [26]. In the case of China, the growth in the number of large-scale industrial enterprises has been constrained by a lack of freshwater resources [27, 28]. Crow-Miller noted that central government leaders viewed the South-to-North Water Diversion Project as a tool to promote economic growth, to some extent supporting the idea that water scarcity has limited the growth of large-scale industrial enterprises in the northern regions [29, 30]. Based on this theory, after the South-to-North Water Diversion Project provides the northern regions with a more abundant water supply, there should be a noticeable additional increase in the number of large-scale industrial enterprises in the beneficiary regions once the project becomes operational and changes the water supply situation compared to the past.

Based on the above, this paper proposes hypothesis 1: The Middle Route of the SNWDP has contributed to an additional growth in the number of IEDS in the beneficiary areas.

## The equilibrium and equilibrating effects of large-scale water infrastructure

In the perspective of strategic management, mixed-ownership reforms in state-owned enterprises can stimulate the intrinsic vitality of micro-level entities, continuously improve their business practices, foster innovation, enhance management efficiency, and increase capital returns [31–33]. One of the challenges in enterprise ownership reform lies in the post-reform confidence of some enterprises in obtaining resources without the prior policy support. The South-to-North Water Diversion Project directly increased the resource supply, effectively addressing the water resource-related challenges that some enterprises faced during the reform process [34].

Based on the economic benefits resulting from differences in water resource access in a market scenario, we now propose Hypothesis 2A: The South-to-North Water Diversion Project promotes the growth of large-scale enterprises in beneficiary regions, with differences existing among enterprises of different ownership structures.

The costeffectiveness of the SNWDP and the value added of water resources are both dependent, at least in part, on the allocation of water resources within the basin, according to a recent analysis by Fang et al [35]. that synthesized all the aforementioned costs and benefits. The assumption of a balanced distribution of transferred water resources in the basin is made because the transferred water of the Middle Route of SNWDP is primarily used for backfilling groundwater and domestic production water supply with strong directivity, and because the regional environmental compensation effect of the four provinces and cities calculated according to the existing calculation is essentially the same [36, 37]. Hypothesis 2B: When the Middle Route of SNWDP promotes the growth of IEDS in beneficiary areas, there is no regional difference for regions with different development levels.

## The general factors of economic growth that promote the increase in the number of IEDS in northern China

According to Cobb-Douglas function, the development level of enterprises is related to capital input and labor input [38]. According to Bartelsman and Doms, with the gradual

improvement of firm-level data, the concept of total factor productivity can be extended to the enterprise level [39]. Total factor productivity (TFP) refers to the growth rate of additional output of enterprises obtained through the improvement of technological level, the improvement of resource utilization efficiency and the influence of policies without increasing factor input at the enterprise level [40]. It can be seen that technical factors and resource factors may lead to the growth of additional output of enterprises [41]. In this way, for the factors that promote the production growth of IEDS, we think that they may be composed of labor force factor, capital factor, technology factor and resource factor through synthesizing two viewpoints.

In order to explore the influence mechanism of the Middle Route of SNWDP on the number growth of IEDS, this paper puts forward the following hypotheses by synthesizing the above theoretical contents related to enterprise development: The increase in industrial businesses over the designated size brought on by the SNWDP's water supply is mostly due to the alteration of the natural endowment. Therefore, this paper proposes hypothesis 3A: The mechanism that the SNWDP increases the number of IEDS in the beneficiary areas is related to the increase of water supply; Hypothesis 3B: The mechanism of the South-to-North Water Diversion to increase the number of IEDS in the beneficiary areas has nothing to do with capital accumulation; Hypothesis 3C: The mechanism of the South-to-North Water Diversion Project to increase the number of IEDS in the beneficiary areas has nothing to do with the development of technology market. Because the increase of population can bring consumption and labor force to the region, and then promote the development of regional industry, this paper puts forward the hypothesis 3D: The mechanism by which the South-to-North Water Diversion Project increases the number of IEDS in the beneficiary areas is related to the inflow of population.

## Experimental design and data description

### Introduction of the middle route project of South-to-North Water Diversion Project

China can be said to be a typical example of water supply and demand disconnect [42]. China may have ample water resources nationwide, but the majority of them are concentrated in the Yangtze River basin, which is too hilly to support China's largest needs for large-scale agriculture and industry [43]. On contrast, there is a significant per capita water deficit on the North China Plain, which is home to many of China's most productive agricultural centers and sizable urban districts [44]. Water shortages are made worse on the North China Plain by the fact that many rivers there have low water quality [45]. The scarcity has been made worse by the North China Plain's significance to China's food supply and economy [4]. The water pressure in the area has been a concern for China's successive governments. Chairman Mao Zedong's SNWDP is China's major effort to address the North China Plain's water shortage through the largest inter-basin water transfer in human history.

A massive and far-reaching inter-basin water diversion project, the Middle Route of SNWDP is now under construction. The main canal, with a total length of about 1276 km, diverts water from the Taocha canal of the Danjiangkou reservoir through Hubei, Henan, Hebei, Tianjin, Beijing, and other provinces and cities (municipalities directly under the Central Government), benefiting about 79 million people in Henan, Hebei, Tianjin, and Beijing. In Phase I of the Middle route Project, 9.5 billion m3 of water are expected to be diverted. The original balance of water resources and ecosystem has gradually changed since the project's first phase's water supply was completed in 2014, which has a significant impact on the cities, groundwater, ecological environment, industrial structure, and social development of the water transfer area and the receiving area [46].

The year 2014 is used as the research object in this study, and the Middle Route of the SNWDP's corresponding time point serves as the study's anchor for observations. In order to understand the promotion effect of the Middle Route of SNWDP on the number growth of IEDS in the beneficiary areas, the treatment effect of this project on the number growth of IEDS is studied using a difference-in-difference model.

## Variable definition

**Water supply volume and number of IEDS.** Due to the differences in economic development levels and stages between northern and southern regions, conducting a direct analysis of growth figures and proportions would undoubtedly entail significant endogeneity issues. Therefore, by differencing the data of the beneficiary regions and the data from other regions in the country, and removing the individual effects of the beneficiary regions, we can obtain the net policy effects for comparison.

The explanatory variable in this study is water supply volume, sourced from the "China Water Resources Bulletin" with the corresponding statistical table named "Provincial-level Administrative Region Water Supply and Water Usage" and the specific item being "Total Water Supply (100 million cubic meters)". The dependent variable is the number of IEDS, obtained from the "China Statistical Yearbook (2010 2018)" with the relevant statistical table named "Main Indicators of Large and Medium-sized Industrial Enterprises in Various Regions" and the specific item being "Number of Enterprise Units (count)". This study synthesizes the data of 31 provincial administrative regions published in China Statistical Yearbook (2010–2018) in 9 years, and obtains 279 samples, which together with DID core explanatory variables and control variables constitute a strong balance panel.

**Control variable selection.** The control variables in benchmark analysis, heterogeneity analysis and robustness analysis are mainly composed of factor variables and water resources variables. The factor variable group includes total capital formation (100 million yuan), technology market turnover (10,000 yuan), and year-end population (10,000 people) to ensure that our model can control the differences in labor, technology and capital among provinces. The water variable group includes water supply (100 million cubic meters) and groundwater resources (100 million cubic meters), which helps us to control for differences in water supply due to changes in water allocation between provinces and cities.

In the mechanism analysis, we add the firm number variable group as a supplementary control variable. The variable group of number of companies includes the number of state-owned and state-controlled industrial enterprises (SIE) (number), the number of private industrial enterprises (PIE) (number) and the number of foreign-invested and Hong Kong-Macao-Taiwan-invested industrial enterprises (FHMTIE) (number), which can help us control the differences in the use of different production factors in different provinces under different enterprise numbers.

The above three groups of control variables are all from China Statistical Yearbook (2010 2018). This study synthesizes the contents published by 31 provincial administrative regions in China Statistical Yearbook (2010 2018) in 9 years, and obtains 2232 data, which together with the explained variables and DID core explanatory variables constitute a strong balance panel.

## Descriptive statistics

Before conducting a formal analysis, we provide the following descriptive statistics. Based on the descriptive statistics in Table 1, it can be observed that the variables of China's technology market transaction volume, the number of large-scale industrial enterprises, the number of private enterprises, and total capital formation exhibit standard deviations above 5, indicating

**Table 1. Descriptive statistics.**

| Variable | N | Mean | p50 | SD | Min | Max |
|---|---|---|---|---|---|---|
| IEDS (number)/1000 | 279 | 12.45 | 6.426 | 14.17 | 0.0560 | 66.31 |
| PIE (number)/1000 | 279 | 7.519 | 3.538 | 9.346 | 0.00800 | 46.71 |
| SIE (number)/1000 | 279 | 0.637 | 0.647 | 0.306 | 0.0250 | 1.935 |
| FHMTIE (number)/1000 | 279 | 1.885 | 0.531 | 3.445 | 0.00300 | 19.19 |
| Population at the end of the year (10,000 persons)/1,000 | 279 | 4.423 | 3.826 | 2.813 | 0.296 | 12.68 |
| Technical market turnover (ten thousand yuan)/100,000 | 279 | 42.37 | 10.07 | 86.73 | 0.00400 | 700.6 |
| Gross capital formation (100 million yuan)/1000 | 279 | 11.14 | 9.551 | 7.893 | 0.381 | 39.66 |
| Water supply (100 million cubic meters)/1000 | 279 | 0.195 | 0.181 | 0.143 | 0.0220 | 0.619 |
| Groundwater (100 million m$^3$)/1000 | 279 | 0.261 | 0.204 | 0.219 | 0.00400 | 1.106 |

significant variability around the mean values. However, considering the adoption of provincial-level data in this study and the uneven development of private economy and capital markets across different Chinese provinces, this volatility precisely reflects the real-life situation and assists us in better conducting the subsequent mechanism analysis.

The total number of samples for each variable is the same. Therefore, according to the results of descriptive statistics, the variables selected this time can be adopted.

## Model setting

Our empirical strategy follows the standard DID approach [47]. We compare the relative changes in the number of IEDS between the beneficiary and nonbeneficiary areas of the Middle Route of SNWDP. The model is constructed as follows:

$$\text{Big}_{pt} = \beta \ \text{Treated}_p \times \text{After}_t + \delta_p + \sigma_t + x_{pt} + \varepsilon_{pt} \tag{1}$$

Where $p$ is the province and $t$ is the year; $\text{Big}_{pt}$ refers to the number of IEDS; $\text{Treated}_p$ is a dummy variable, if the province is the beneficiary area of the Middle Route of SNWDP, equal to 1, otherwise equal to 0. The treatment group included Beijing, Tianjin, Hebei and Henan, while the control group included 27 provincial-level administrative regions of the 31 provincial-level administrative regions that were not part of the four beneficiary regions. We take $\text{After}_t$ as dummy variables, the year after the water supply of the Middle Route of SNWDP (2014) is equal to 1, and the previous year is 0. The equation also includes the control of fixed effects $\delta p$ and $\sigma t$ of provincial administrative region and year; $\chi ct$ represents the control at the time of the change in policy. The correlation coefficient in Eq (1) is $\beta$, which is the influence of the water supply of the Middle Route of SNWDP on the number growth of IEDS. We expect this coefficient to be positive, which will indicate that the water supply of the Middle Route of SNWDP will promote the growth of the number of IEDS.

This estimation strategy has the advantages and potential disadvantages of the classical DID model. The provincial-level fixed effect controls for all interprovincial individual variables. Time fixed effects control the factors that promote the growth of industrial enterprises in all provinces and cities. Our model construction relies on the assumption that there are no other missing variables or events that are consistent with economic factors or affect the growth of the number of IEDS, except for the variables we control. This assumption is flawed because there are many factors (including industrial cluster effect, regional policy factors, etc.) that affect the growth of the number of IEDS. We will discuss this issue in Robustness Statistics.

## Empirical results and analysis

### Benchmark analysis

**Parallel trend test.** The premise of difference-in-difference method to estimate the effectiveness of policy is that the experimental group and the control group have similar growth or decroute trend before the policy shock, so it is necessary to carry out parallel trend test on the explained variable. In this paper, a dummy variable is set for each year before the implementation of the Middle Route of SNWDP, and multiplied with the dummy variables of the experimental group. Then the five dummy variables and the DID core explanatory variables $\text{Treated}_p \times \text{After}_t$ are used to regress the industrial structure level coefficient and industrial structure height. The results are shown in Fig 1, where the DID core explanatory variables are significantly positive, and the dummy variable coefficients multiplied by the years before 2014 and the experimental group are not significant. In this way, it basically proves that the model satisfies the parallel trend assumption. The findings reveal that before the Middle Route of SNWDP was put into place, the average growth trend for the number of IEDS was the same in both the experimental group and the control group, while the growth rate of the experimental group was higher than that of the control group after the proposal was made. Therefore, it can be judged that the empirical model satisfies the parallel trend hypothesis.

**Mean effect test.** This paper adopts two-way fixed effect model, fixed regional effect and time effect to study the influence of SNWDP on the number growth of IEDS. Table 2 reports the baseroute regression results for the mean effect test. For model (1), we added factor variables to account for the differences in key factors of production-labor, technology, and capital-

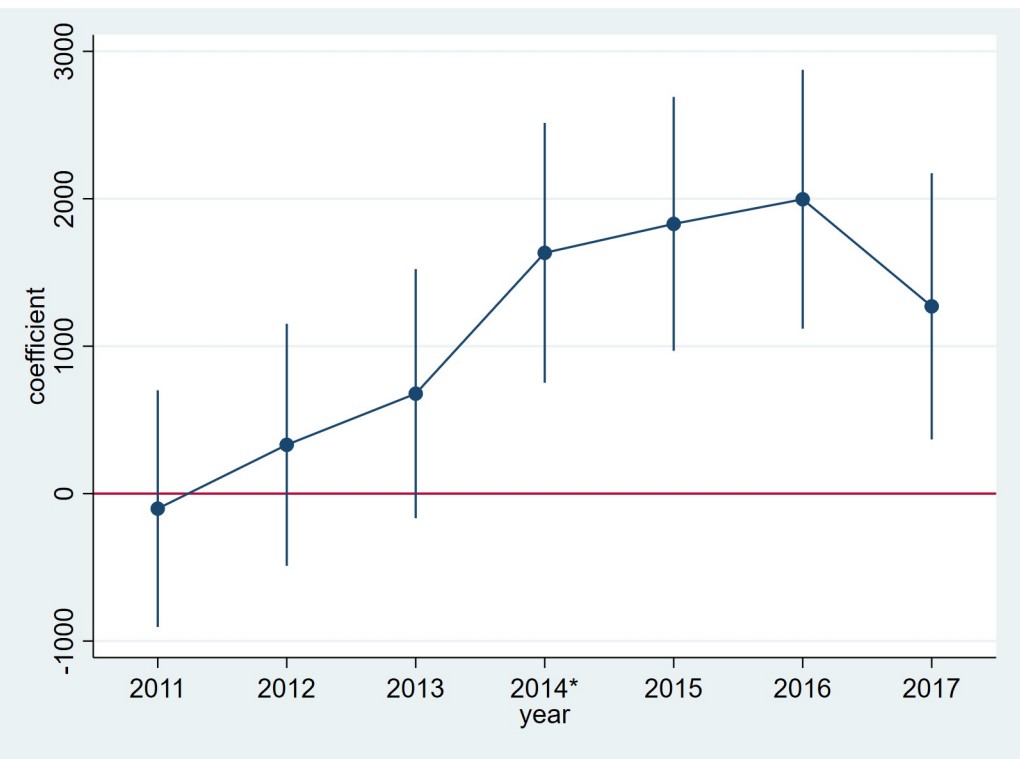

**Fig 1. Parallel trend test of the number of IEDS in the experimental group and the control group before and after water supply.**

**Table 2. Average effect test of water supply in the middle route of SNWDP and the number of IEDS: Benchmark regression.**

| | (1) | (2) | (3) |
|---|---|---|---|
| | *IEDS(number)* | *IEDS(number)* | *IEDS(number)* |
| $Treated_p \times After_t$ | 961.3*** | 935.7*** | 963.9*** |
| | (253.4) | (251.3) | (249.5) |
| Factor variable | Yes | Yes | Yes |
| Regional effect | No | Yes | Yes |
| Time effect | No | Yes | Yes |
| Water resources variable | No | No | Yes |
| _cons | 23723.5*** | 22853.5*** | 19257.3*** |
| | (4519.1) | (4532.5) | (4754.1) |
| Dependent variable mean | 12263.53 | 12263.53 | 12263.53 |
| N | 241 | 241 | 241 |
| $R^2$ | 0.955 | 0.957 | 0.959 |
| adj.$R^2$ | 0.947 | 0.948 | 0.949 |

\* Note: [1] Standard errors in parentheses \*$p < 0.1$, \*\*$p < 0.05$, \*\*\*$p < 0.01$

[2] The DID core variables in this table are $Treated_p \times After_t$. Where, when the observation site is Beijing, Tianjin, Hebei and Henan, $Treated_p = 1$; when the observation site is the other 31 provincial-level administrative regions where data are published in the statistical yearbook, $Treated_p = 0$. $After_t$ is a dummy variable of whether the policy is implemented or not.

between the regions that benefited from the SNWDP and those that did not. For model (2), we add time effect and region effect to model (1) to consider the different individual effect and time effect between control group and experimental group. In order to cover the situation of all parts of China as much as possible, this study refers to the data published in the Statistical Yearbook, and unifies with it, taking the province as the investigation unit, and the basic unit of all regional effects is the province. Model (3) is based on model (2), adding water resources variable to consider the difference of water resources supply caused by water resources flow between different provinces and cities. The R2 of models (1) to (3) increases gradually, which indicates that the addition of each factor enhanced the explanatory power of the model.

The findings demonstrate that our findings are meaningful, which suggests that when the Middle Route of the SNWDP began supplying water, the number of industrial companies exceeding the designated size increased. For instance, the point estimate shown in column 3 is 963.9, which means that following the connection of the middle route of the SNWDP, the number of industrial firms in the recipient provinces larger than the required size grew by 963.9. This effect is equivalent to a 7.86% increase in the sample mean of 12263.53 and is significant at the 1% level. The estimated coefficients reported in columns 1 to 2 show similar magnitudes and significance levels as those reported in column 1.

The SNWDP has greatly expanded the number of industrial companies larger than the required size, demonstrating the validity of hypothesis 1.

### Heterogeneity analysis

**Possible influence of SNWDP on the reform of enterprise ownership.** One potential contributing aspect is that the SNWDP's implementation has aided in the economic growth of the beneficiary areas, which in turn has aided in the mixed ownership reform of SIE. The abundance of water resources has alleviated some resource concerns in enterprises undergoing reforms, extending the impact of reforms to more northern region businesses. Under the

**Table 3. Mechanism analysis of the promotion effect of water supply in the middle route of SNWDP on the number of IEDS: Enterprise ownership.**

| | (1) | (2) | (3) | (4) | (5) | (6) |
|---|---|---|---|---|---|---|
| | *PIE(number)* | *PIE(number)* | *FHMTIE(number)* | *FHMTIE(number)* | *SIE(number)* | *SIE(number)* |
| $Treated_p \times After_t$ | -595.7 | -492.8 | 144.3 | 184.3 | -66.05** | -63.72** |
| | (547.2) | (619.9) | (145.7) | (139.8) | (24.61) | (26.66) |
| factor variable | Yes | Yes | Yes | Yes | Yes | Yes |
| regional effect | Yes | Yes | Yes | Yes | Yes | Yes |
| time effect | Yes | Yes | Yes | Yes | Yes | Yes |
| water resources variable | No | Yes | No | Yes | No | Yes |
| _cons | -42933.1*** | -47582.7*** | 5142.8 | 2068.4 | 1314.7 | 1060.5 |
| | (12933.4) | (12697.7) | (4870.7) | (5132.3) | (923.6) | (960.0) |
| Dependent variable mean | 7003.15 | 7003.15 | 1884.91 | 1884.91 | 609.28 | 609.28 |
| $N$ | 241 | 241 | 241 | 241 | 241 | 241 |
| $R^2$ | 0.335 | 0.348 | 0.558 | 0.592 | 0.420 | 0.434 |
| adj. $R^2$ | 0.300 | 0.307 | 0.534 | 0.567 | 0.390 | 0.399 |

* Note: [1] Standard errors in parentheses $*p < 0.1$, $**p < 0.05$, $***p < 0.01$

[2] The DID core variables in this table are $Treated_p \times After_t$. Where, when the observation site is Beijing, Tianjin, Hebei and Henan, $Treated_p = 1$; when the observation site is the other 31 provincial-level administrative regions where data are published in the statistical yearbook, $Treated_p = 0$. $After_t$ is a dummy variable of whether the policy is implemented or not.

reform, the improvement of enterprise efficiency has brought about the increase of main business income of enterprises, so that more enterprises have crossed the threshold of main business income of 20 million yuan, and the number of IEDS has increased.

The average impact of the water supply from the Middle Route Project of the SNWDP on the growth of the number of businesses with various ownership is tested, as shown in Table 3. Columns (1) (4) show the relationship between the number of PIE, the number of FIE and the core variables of DID. The findings indicate that the data we obtained are not statistically significant, which suggests that the Middle Route Project of the SNWDP's water supply has a substantial impact on changes in the number of private firms and FIE. Columns (5)-(6) show the relationship between the number of SIEs and the core variables of DID. The results show that all negative coefficients of the results we obtained are significant, and the point estimate in (6) is -63.72, accounting for 10.46% of the dependent variable mean. It demonstrates how the Middle Route of the SNWDP's water supply has resulted in a decrease in the number of state-owned businesses and state-owned holding companies.

Therefore, assuming 3A is established, when the South-to-North Water Diversion Project promotes the growth of the number of IEDS in the beneficiary areas, there are differences for enterprises with different ownership.

Only state-owned, privately owned, and foreign-funded businesses make up the majority of Chinese enterprises today, with the number of state-owned businesses declining significantly even without the Middle Route of SNWDP's effects. Given these facts and the fact that China's state-owned businesses rarely go bankrupt, it is possible to conclude that the SNWDP can help advance the implementation of mixed ownership reform because the Middle Route Project's water supply has partially changed the ownership of state-owned businesses. Hypothesis 2A holds.

**Possible influence of SNWDP on regional economic equilibrium.** Marshall believes that the reason why enterprises initially gather in a certain place is natural endowment and policy orientation, that is, regions with better natural endowment and policy orientation are easy to

**Table 4. Average effect test of water supply in the middle route of SNWDP and the number of IEDS: Benchmark regression.**

|  | (1) | (2) | (3) |
|---|---|---|---|
|  | *IEDS(number)* | *IEDS(number)* | *IEDS(number)* |
| $Treated_p \times After_t \times Beijing$ | -1315.9 | -960.6* | -883.1 |
|  | (832.4) | (535.2) | (529.9) |
| Factor effect | Yes | Yes | Yes |
| Regional effect | No | Yes | Yes |
| Time effect | No | Yes | Yes |
| Water resources effect | No | No | Yes |
| _cons | 59970.4*** | 79312.0*** | 89540.0*** |
|  | (10025.7) | (22170.5) | (28454.2) |
| N | 32 | 32 | 32 |
| $R^2$ | 0.879 | 0.967 | 0.974 |
| adj. $R^2$ | 0.821 | 0.932 | 0.937 |

* Note: [1] Standard errors in parentheses *$p < 0.1$, **$p < 0.05$, ***$p < 0.01$

[2] The DID core variables in this table are $Treated_p \times After_t \times Beijing_p$. Where, when the observation site is Beijing, Tianjin, Hebei and Henan, $Treated_p = 1$; when the observation site is the other 31 provincial-level administrative regions where data are published in the statistical yearbook, $Treated_p = 0$. $After_t$ is a dummy variable of whether the policy is implemented or not. When the observation site is Beijing, $Beijing_p = 1$; When the observation site is Tianjin, Hebei and Henan, $Beijing_p = 0$.

breed advantageous enterprises [26]. The results in the baseroute regression are the average effects of the study on the beneficiary areas as a whole, which does not take into account the uneven development of provinces within the beneficiary areas. However, since the transferred water is mainly used for backfilling groundwater and domestic production water supply with strong orientation, this paper believes that the distribution of transferred water resources within the beneficiary areas is relatively fair, which will not cause uneven growth of IEDS within the beneficiary areas.

In Table 4, the policy benefit difference within the income area of the Middle Route of SNWPT is studied using a two-way fixed effect model, whether to add Beijing dummy variable on the basis of the original interaction term, and a heterogeneous difference-in-difference model. To analyze the impact of the SNWDP on the expansion of industrial firms over a specified size, we fixed the regional effect and temporal effect in the model. Table 4 reports the results of the mean effect test. It turns out that the results we get are basically insignificant. This shows that after the middle route of the SNWDP is connected to the water, there is no inequality in the growth of the number of IEDS within the beneficiary provinces in the Huang-Huai-Hai basin.

As a result, validity of Hypothesis 2B can be demonstrated. The SNWDP's water supply has the same encouraging effect on the expansion of industrial businesses above a predetermined size in the four provinces, despite the fact that Beijing's economic development is significantly more advanced than that of the other provinces.

## Robustness analysis

**Contemporaneous economic disturbance analysis.** After examining how the middle route of the South-to-North water transfer affects the expansion of industrial businesses above a certain size, we can say that it encourages expansion of IEDS. The SNWDP's allocation of water resources may have different effects in different regions due to differences in economic status, economic development stages, and industrial water demand levels, which are not taken into account in the aforementioned results, which represent the average effects of the study at

**Table 5. Average effect test of water supply in the middle route of SNWDP and the number of IEDS: Contrast North and South.**

| | (1) | (2) | (3) |
|---|---|---|---|
| | *IEDS(number)* | *IEDS(number)* | *IEDS(number)* |
| $Group_p \times After_t$ | 1645.6*** | 1581.2*** | 1378.7*** |
| | (426.9) | (405.2) | (393.8) |
| Factor variable | Yes | Yes | Yes |
| Regional effect | No | Yes | Yes |
| Time effect | No | Yes | Yes |
| Water resources variable | No | No | Yes |
| _cons | 53564.5*** | 30287.8* | 33131.2** |
| | (10728.1) | (15633.4) | (14555.2) |
| Dependent variable mean | 20798.65 | 20798.65 | 20798.65 |
| N | 64 | 64 | 64 |
| $R^2$ | 0.977 | 0.983 | 0.986 |
| adj.$R^2$ | 0.971 | 0.975 | 0.978 |

Note: [1] Standard errors in parentheses *$p < 0.1$, **$p < 0.05$, ***$p < 0.01$.

[2] The DID core variables in this table are $Group_p \times After_t$. Where, when the observation site is Beijing, Tianjin, Hebei and Henan, $Group_p = 1$; When the observation site is Shanghai, Jiangsu, Zhejiang and Anhui, $Group_p = 0$. $After_t$ is a dummy variable of whether the policy is implemented or not.

the level of 31 provinces in China. At the same time, it does not consider whether other economic policies have taken effect during the same period, which has made enterprises in more developed areas have greater development. Similar economic position, similar economic linkages across provinces, and similar business environments can be found in the Yangtze River Delta region as well as the hinterland of North and Central China, including Beijing, Tianjin, Hebei, and Henan. Therefore, it was considered as a control group and compared with the experimental group separately.

This comparative study adopts two-way fixed effect model, fixed regional effect and time effect, narrowed the scope of comparison group, focused on the Yangtze River Delta region (Shanghai, Zhejiang, Jiangsu, Anhui) with similar economic conditions to the beneficiary regions (Beijing, Tianjin, Hebei and Henan), observing the impact of the SNWDP on the number growth of IEDS, so as to exclude the possible impact of other macroeconomic factors at the same time. Table 5 reports the regression test results for the mean effect test, and for several models we added the control variables and effects in the same order as in Table 2.

The findings demonstrate that our findings are significant, which suggests that the Middle Route of the SNWDP's water supply has encouraged the expansion of industrial companies beyond the required scale. For instance, the point estimate shown in column 3 is 1378.7, which indicates an additional increase of 1378.7 in the number of industrial firms in the beneficiary provinces that are larger than the designated size after receiving water from the Middle Route of the SNWDP. This effect corresponds to 6.63% of the sample mean of 20798.65 which is significant at the 1% level. The estimated coefficients reported in columns 1 to 2 show similar magnitudes and significance levels as those reported in column 3. The fact that this number is not significantly different from that of China's 31 provinces suggests that other economic changes during the same time period did not significantly affect the SNWDP's ability to significantly increase the number of IEDS.

The results of assumption 1 are robust.

**Placebo test.** In order to test whether the increase in the number of IEDS is the growth effect brought by time change, and to exclude the unobserved influence of the sample

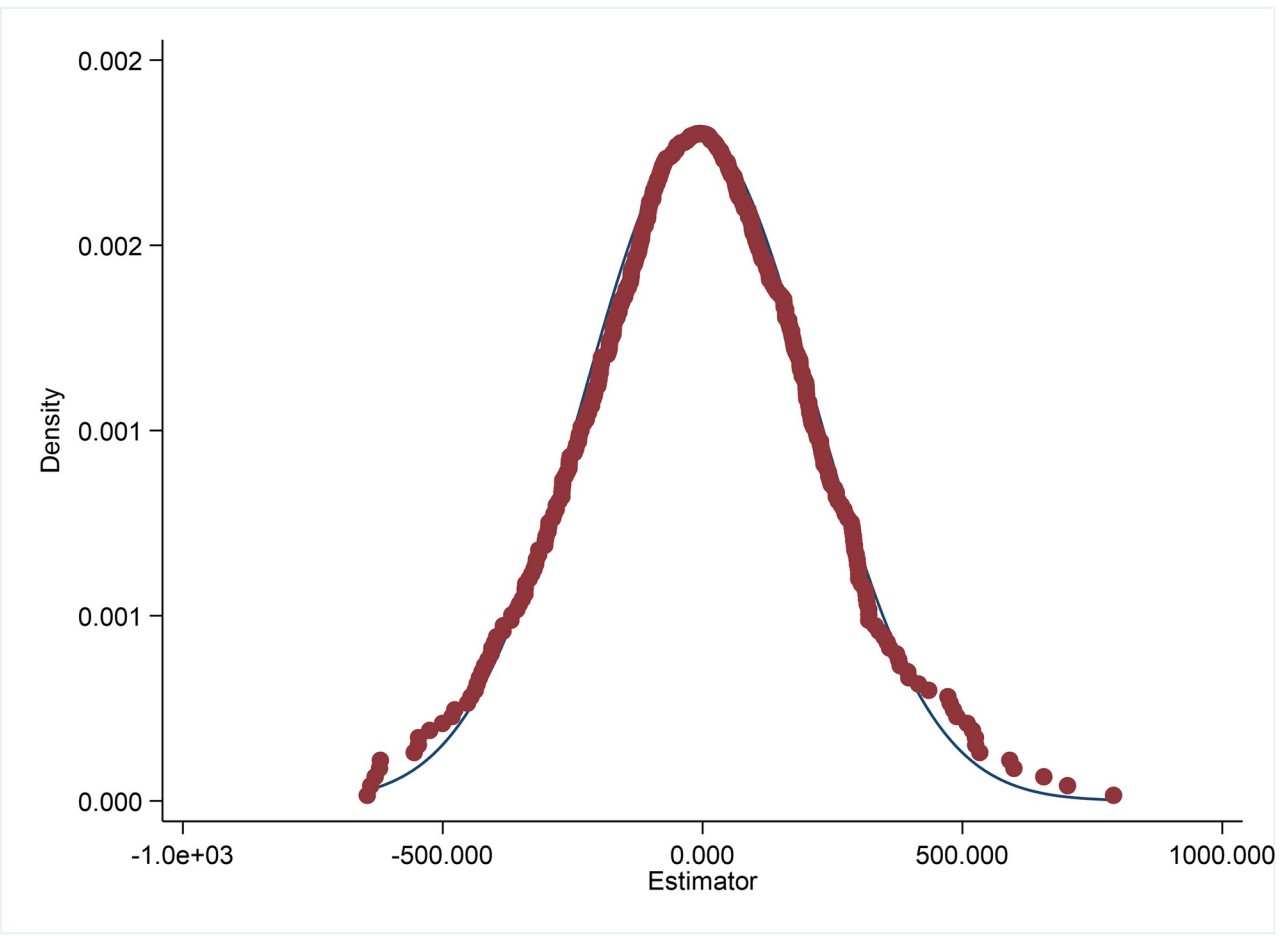

**Fig 2. Coefficient distribution of quantity coefficient of IEDS after random sampling.**

characteristics of provincial administrative regions on the regression results, 143 samples were randomly selected from all 403 samples as "pseudo-experimental group" for placebo test, and the random sampling process was repeated 500 times, and the product of time dummy variable and time dummy variable was adopted as the core explanatory variable for regression. Fig 2 shows the coefficient distribution of the regression results of the number of IEDS as the explanatory variable. It can be seen from the figure that the regression coefficient distribution of the two models is concentrated around 0 and has normal distribution characteristics, which shows that the sample combination after random sampling has no effect on the number of IEDS. Therefore, it can be concluded that the regression results of the benchmark regression by whether the provincial administrative regions benefit from the Middle Route Project of South-to-North Water Diversion between the experimental group and the control group are robust.

**Change policy impact time.** Through the previous study, we found that the water supply of the middle route of SNWDP has a certain role in promoting the number of IEDS, but it may be the result of other policies or reasons. In order to further prove the robustness of the empirical results, this paper adopts the method of counterfactual test to investigate whether the core explanatory variables are still significant when the Middle Route of SNWDP is not

**Table 6. Average effect test of change policy impact time.**

|  | (1) | (2) | (3) |
|---|---|---|---|
|  | *IEDS(number)* | *IEDS(number)* | *IEDS(number)* |
| $Treated_p \times After\_1_t$ | 909.5 | 906.1 | 975.8 |
|  | (740.1) | (715.9) | (711.4) |
| Factor effect | Yes | Yes | Yes |
| Regional effect | No | Yes | Yes |
| Time effect | No | Yes | Yes |
| Water resources effect | No | No | Yes |
| _cons | 22529.8*** | 23717.4*** | 20024.3** |
|  | (6817.7) | (7700.9) | (8359.6) |
| $N$ | 241 | 241 | 241 |
| $R^2$ | 0.954 | 0.957 | 0.959 |
| adj.$R^2$ | 0.953 | 0.955 | 0.956 |

Note: [1] Standard errors in parentheses *$p < 0.1$, **$p < 0.05$, ***$p < 0.01$.

[2] The DID core variables in this table are $Treated_p \times After\_1_t$. Where, when the observation site is Beijing, Tianjin, Hebei and Henan, $Treated_p = 1$; when the observation site is the other 31 provincial-level administrative regions where data are published in the statistical yearbook, $Treated_p = 0$. $After\_1_t$ is the dummy variable of whether the counterfactual policy is implemented or not, when the observation time is in 2013 and before, $After\_1_t = 0$; When the observation time is after 2013, $After\_1_t = 1$.

connected. If it is significant, it indicates that there are other unobserved factors that promote the growth of the number of IEDS. If it is not significant, it indicates that the promotion effect of the water supply of the Middle Route of SNWDP on the growth of the number of IEDS is stable and reliable. In this paper, the year before the water supply of the Middle Route of SNWDP, i.e. 2013, is adopted as the core explanatory variable for the test, and then the regression is carried out again. The results are shown in Table 6. The regression results are not significant, so it can be judged that the model basically conforms to the counterfactual hypothesis, which shows that the promotion effect of the water supply of the Middle Route of SNWDP on the number of IEDS is robust.

## Mechanism analysis

The earlier parts present direct causal evidence connecting the increase in the number of industrial companies larger than the designated size to the water supply of the Middle Route of the SNWDP. This section discusses the potential mechanisms by which the Middle Route of the SNWDP promotes the growth of IEDS. We consider five possible factors: labor, capital, technology, ownership and resources.

**Changes of regional water supply after the policy of SNWPT.** As stated in the introduction and policy introduction, China's water supply and demand were significantly out of balance prior to the SNWDP, and the lack of water resources severely hampered the environmental, economic, and social growth of the northern region. After the Middle Route of SNWDP is connected, the water supply in the beneficiary areas increases, resulting in the optimization of the natural endowment of the beneficiary areas, which can organize larger-scale production activities or improve labor productivity. This is another potential explanation for the significant increase in the number of IEDS.

As can be seen from Table 7, the results obtained are all significant, which indicates that the water supply of the Middle Route Project of SNWDP promotes the increase of water supply in the beneficiary areas. For example, the point estimate reported in column 3 is 6.965, which represents an increase of 696.5 million m3 in water supply in the beneficiary provinces after

**Table 7. Average effect test of change policy impact time.**

| | (1) | (2) | (3) |
|---|---|---|---|
| | Water supply (100 million cubic meters) | Water supply (100 million cubic meters) | Water supply (100 million cubic meters) |
| $Treated_p \times After\_1_t$ | 6.891** | 6.690* | 6.965** |
| | (3.461) | (3.466) | (3.429) |
| Firm quantity variable | Yes | Yes | Yes |
| Factor effect | No | Yes | Yes |
| Regional effect | No | No | Yes |
| Time effect | No | No | Yes |
| _cons | 183.2*** | 273.9** | 170.0 |
| | (9.375) | (105.2) | (238.7) |
| Dependent variable mean | 195.76 | 195.76 | 195.76 |
| N | 72 | 64 | 64 |
| $R^2$ | 0.219 | 0.228 | 0.366 |
| adj.$R^2$ | 0.060 | 0.008 | 0.002 |

Note: [1] Standard errors in parentheses $*p < 0.1$, $**p < 0.05$, $***p < 0.01$.

[2] The DID core variables in this table are $Treated_p \times After_t$. Where, when the observation site is Beijing, Tianjin, Hebei and Henan, $Treated_p = 1$; when the observation site is the other 31 provincial-level administrative regions where data are published in the statistical yearbook, $Treated_p = 0$. $After_t$ is a dummy variable of whether the policy is implemented or not.

the Middle Route of the SNWDP is connected. This effect is equivalent to 3.56% of the sample mean of 19.576 billion cubic meters and is significant at the 5% level. The estimated coefficients reported in columns 1 to 2 show similar magnitudes and significance levels as those reported in column 3. This value proves that the water transferred from the Middle Route of SNWDP not only plays a role in backfilling groundwater, but also provides necessary water resources guarantee for production and life, which can further explain the promotion effect of SNWDP on industry.

Therefore, assuming that 3B is established, the mechanism of the increase in the number of IEDS in the beneficiary areas of the South-to-North Water Diversion Project is related to the increase in water supply.

**Changes of regional capital elements after the policy of SNWDP.** Table 8 shows that all of our results are significant negative, proving that the Middle Route of the SNWDP's water supply did not boost the overall capital formation in the beneficiary areas. For example, the point estimate reported in column 3 is -1209.0, which means that the total capital formation of the beneficiary provinces after the water connection of the Middle Route of the SNWDP is 120.9 billion yuan less than the amount that would have been expected under a parallel trend with other provinces in the same period. This effect is equivalent to 10.9% of the sample mean of 1114.223 billion yuan, which is significant at the 1% level. The estimated coefficients reported in columns 1 to 2 show similar magnitudes and significance levels as those reported in column 3. From the observation of the raw data, it can be seen that the total capital formation has maintained an upward trend in the observed years, which indicates that the negative coefficient here means that the total capital formation of the beneficiary areas has not reached the same growth rate as that of other provinces in the country during the observed years.

**Table 8. Mechanism analysis of the promotion effect of water supply in the middle route of SNWDP on the number of IEDS: Gross capital formation.**

|  | (1) | (2) | (3) |
|---|---|---|---|
|  | *Gross capital formation (100 million yuan)* | *Gross capital formation (100 million yuan)* | *Gross capital formation (100 million yuan)* |
| $Treated_p \times After_t$ | -1435.2*** | -1660.4*** | -1209.0*** |
|  | (444.6) | (430.0) | (407.6) |
| Firm quantity variable | Yes | Yes | Yes |
| Water resources variable | No | Yes | Yes |
| Regional effect | No | No | Yes |
| Time effect | No | No | Yes |
| _cons | -4699.4 | -37389.5*** | -30735.3*** |
|  | (6073.3) | (8568.3) | (8200.8) |
| Dependent variable mean | 11142.23 | 11142.23 | 11142.23 |
| N | 241 | 241 | 241 |
| $R^2$ | 0.780 | 0.799 | 0.824 |
| adj.$R^2$ | 0.739 | 0.759 | 0.782 |

Note: [1] Standard errors in parentheses *$p < 0.1$, **$p < 0.05$, ***$p < 0.01$.

[2] The DID core variables in this table are $Treated_p \times After_t$. Where, when the observation site is Beijing, Tianjin, Hebei and Henan, $Treated_p$ = 1; when the observation site is the other 31 provincial-level administrative regions where data are published in the statistical yearbook, $Treated_p$ = 0. $After_t$ is a dummy variable of whether the policy is implemented or not.

Gross capital formation is the sum of gross fixed capital formation, the value of changes in inventory, and the acquisition of valuable items, less the value of disposals. There are three possible reasons for this phenomenon: First, the average total capital formation in the beneficiary areas is higher than the average of 31 provinces in China, and the capital formation rate has the characteristics of marginal decroute, so the growth rate slows down. Second, because some capital was invested in the field of water resources acquisition before the middle route of the SNWDP was connected, some capital investment decreased after the water connection, and the investment section of the total capital formation decreased directionally. Third, it may also be out of the reduction of social fixed capital and inventory. Social fixed capital may include facilities for obtaining water resources, and the reduction of inventory may include some commodities released by water after water supply. Therefore, the inventory section of gross capital formation has a directional reduction.

However, the capital factor is not the main influencing factor of the rise in the number of IEDS brought on by the water supply of the Middle Route of SNWPT because the growth rate of total capital formation fails to reach the same level as that of other regions of the country. Therefore, assuming that 3C is established, the mechanism of the increase in the number of IEDS in the beneficiary areas of the South-to-North Water Diversion Project has nothing to do with capital accumulation.

**Changes of regional technical factors after the policy of SNWPT.** Table 9 shows that all of our results are unfavorable and statistically significant, demonstrating that the Middle Route Project's water supply did not improve the technical market turnover in the project's beneficiary areas. For example, the point estimate reported in column 3 is -832408.9, which means that the technology market turnover of the beneficiary province after the water supply of the Middle Route of SNWDP is 8.324 billion yuan less than the amount that would have been expected under the trend of parallel with other provinces in the same period. This effect is equivalent to 3.25% of the sample mean of 255.986 billion yuan, which is significant at the 5% level. The estimated coefficients reported in columns 1 to 2 show similar magnitudes and

**Table 9. Mechanism analysis of the promotion effect of water supply in the middle route of SNWDP on the number of IEDS: Technology market turnover.**

| | (1) | (2) | (3) |
|---|---|---|---|
| | *Technical market turnover (ten thousand yuan)* | *Technical market turnover (ten thousand yuan)* | *Technical market turnover (ten thousand yuan)* |
| $Treated_p \times After_t$ | -785251.1* | -794324.0** | -832408.9** |
| | (399310.6) | (400282.4) | (402123.0) |
| Firm quantity variable | Yes | Yes | Yes |
| Water resources variable | No | Yes | Yes |
| Regional effect | No | No | Yes |
| Time effect | No | No | Yes |
| _cons | -23994478.9*** | -25546289.4*** | -25598621.0*** |
| | (5141168.3) | (5524625.7) | (6270114.0) |
| Dependent variable mean | 2406163.02 | 2406163.02 | 2406163.02 |
| $N$ | 241 | 241 | 241 |
| $R^2$ | 0.439 | 0.443 | 0.457 |
| adj.$R^2$ | 0.334 | 0.331 | 0.328 |

Note: [1] Standard errors in parentheses *$p < 0.1$, **$p < 0.05$, ***$p < 0.01$.

[2] The DID core variables in this table are $Treated_p \times After_t$. Where, when the observation site is Beijing, Tianjin, Hebei and Henan, $Treated_p = 1$; when the observation site is the other 31 provincial-level administrative regions where data are published in the statistical yearbook, $Treated_p = 0$. $After_t$ is a dummy variable of whether the policy is implemented or not.

significance levels as those reported in column 3. From the observation of the original data, we can see that the technology market turnover has maintained an upward trend in the observed years, which indicates that the negative coefficient here means that the technology market turnover in the beneficiary areas has not reached the same growth rate as other provinces in the country during the observed years.

The turnover in the technology market is the sum of the subject matter of the technology contract (technology development, technology transfer, technology consultation, technical service). The reasons for this phenomenon may therefore be two-fold: First, the average technology market turnover in the beneficiary areas is higher than the average of 31 provinces in the country, and the speed of technology renewal is characterized by marginal decroute, so the growth rate slows down. Second, some technology transactions in the beneficiary areas before the middle route of the SNWDP are aimed at improving the water resource acquisition efficiency or water resource utilization efficiency, and some technology transactions are reduced after the water supply.

However, in any case, because the growth rate of turnover in the technology market fails to reach the same level as that in other parts of the country, the technical factors are not the main influencing factors for the increase of the number of IEDS caused by the water supply of the Middle Route Project of South-to-North Water Diversion. Therefore, assuming that 3D is established, the mechanism by which the South-to-North Water Diversion increases the number of IEDS in the beneficiary areas has nothing to do with the development of technology markets.

**Regional population changes after the SNWPT.** The water supply of the Middle Route Project of the SNWDP has delivered water resources for production and living to northern China, as can be observed from Section 3.4.2. Numerous empirical research findings from around the globe demonstrate a favorable association between population and water resources.

**Table 10. Mechanism analysis of the promotion effect of water supply in the middle route of SNWDP on the number of IEDS: Number of population.**

|  | (1) | (2) | (3) |
|---|---|---|---|
|  | *Population at the end of the year (10,000)* | *Population at the end of the year (10,000)* | *Population at the end of the year (10,000)* |
| $Treated_p \times After_t$ | 39.67** | 39.67** | 42.11** |
|  | (17.62) | (17.62) | (17.72) |
| Firm quantity variable | Yes | Yes | Yes |
| Water resources variable | No | Yes | Yes |
| Regional effect | No | No | Yes |
| Time effect | No | No | Yes |
| _cons | 4341.0*** | 4375.1*** | 4402.0*** |
|  | (35.75) | (83.22) | (86.68) |
| Dependent variable mean | 4373.92 | 4373.92 | 4373.92 |
| N | 241 | 241 | 241 |
| $R^2$ | 0.782 | 0.786 | 0.794 |
| adj.$R^2$ | 0.741 | 0.743 | 0.745 |

Note: [1] Standard errors in parentheses *$p < 0.1$, **$p < 0.05$, ***$p < 0.01$.

[2] The DID core variables in this table are $Treated_p \times After_t$. Where, when the observation site is Beijing, Tianjin, Hebei and Henan, $Treated_p = 1$; when the observation site is the other 31 provincial-level administrative regions where data are published in the statistical yearbook, $Treated_p = 0$. $After_t$ is a dummy variable of whether the policy is implemented or not.

Places with abundant water resources are more likely to attract more people. Population growth can bring consumption and labor to the region, thus promoting local industrial development.

The empirical findings, which are significant under various effect control and variable control, are shown in Table 10 and show that the Middle Route Project of the SNWDP encourages population growth in the recipient districts. The recipient province's population will increase by 421,100 by the end of the year once the middle route of the SNWDP is connected, according to the point estimate provided in column 3 of 42.11. This effect is significant at the 5% level and corresponds to 0.96% of the sample mean of 43,739,200 individuals. Similar magnitudes and levels of significance may be seen in the estimated coefficients presented in columns 1 through 2 as in column 3. This figure indicates that after the middle route of the SNWDP is connected to water, more individuals are willing to opt to work and live in northern China. The SNWDP has enhanced regional prosperity, which in part explains how the recipient areas' economies have grown.

## Conclusions

Large enterprises are important to developing countries, and inter- and in-tra-basin water resources transfers can have a positive effect on the development of IEDS in developing countries.

This essay tries to highlight how the SNWDP's extensive infrastructure can be seen as a chance for economic growth and a solution to the issue of striking a balance between development, balance, and the environment. The Middle Route of the SNWDP and the expansion of industrial businesses past a set size are both investigated in this essay. We present convincing causal evidence that the Middle Route of the SNWDP has contributed to the additional growth of the number of IEDS in the beneficiary areas from the perspective of the difference-in-difference model based on an original dataset covering 9 years of socioeconomic data from 31

provinces. We also discover that the SNWDP's Middle Route has not created any elements that will further encourage the imbalance of regional economic development within the beneficiary areas. Additionally, we discuss how the Middle Route of the SNWDP can support mixed ownership reform in beneficiary areas, increase water supply, and support industrial business expansion above the allowed size. However, the middle route of the SNWDP has had little to no impact on traditional total factor productivity and the production factor components of technology and capital(even though they may have contributed to the growth of IEDS).

The significance of this article lies in demonstrating the positive impact of infrastructure development on economic growth for other developing countries. The successful experience of the Central Route Project of the South-to-North Water Diversion Plan illustrates that strengthening infrastructure can promote the growth of large-scale enterprises, create employment opportunities, and drive economic growth. Other developing countries can learn from this achievement by increasing infrastructure investment to facilitate economic transformation and growth. Additionally, the article emphasizes the importance of mixed-ownership reforms, increased water supply, and population growth in contributing to the growth of industrial enterprises. It provides policy insights and guidance for other countries. In summary, this article offers valuable lessons for other developing countries by highlighting the importance of infrastructure development in promoting sustainable economic development.

This study has some flaws because there is a lack of data. The conclusions may differ from the actual scenario due to the little amount of data used and the simplicity of the composition of the key variables. At the same time, we only examine the effects of the Middle Route Project, and the scientific assessment of the relationship between the SNWDP and industrial development requires the addition of data from the East Route Project to form a multi-period difference-in-difference model for theoretical research and practical evaluation. This is also one of the perspectives for future research.

## Supporting information

**S1 File.**
(XLSX)

## Author Contributions

**Conceptualization:** Ting Wang.

**Data curation:** Ting Wang.

**Formal analysis:** Ting Wang.

**Investigation:** Ting Wang.

**Methodology:** Ting Wang, Jianyu Chi.

**Project administration:** Ting Wang, Jianyu Chi.

**Software:** Ting Wang.

**Supervision:** Jianyu Chi.

**Validation:** Ting Wang.

**Visualization:** Ting Wang.

**Writing – original draft:** Ting Wang, Jianyu Chi.

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
