## [Decision Letter · Decision Letter 0]

24 Jul 2023

PONE-D-23-14222Does the South-to-North Water Transfer Project Promote the Growth of Enterprises above Designated Size in the Water-receiving Areas? - Evidence from 31 provincial-level administrative regions in ChinaPLOS ONE

Dear Dr. Wang,

Thank you for submitting your manuscript to PLOS ONE. After careful consideration, we feel that it has merit but does not fully meet PLOS ONE’s publication criteria as it currently stands. Therefore, we invite you to submit a revised version of the manuscript that addresses the points raised during the review process.

We look forward to receiving your revised manuscript.

Kind regards,

Bing Xue, Ph.D.

Academic Editor

PLOS ONE

Reviewers' comments:

Reviewer's Responses to Questions

**Comments to the Author**

1. Is the manuscript technically sound, and do the data support the conclusions?

Reviewer #1: Yes

2. Has the statistical analysis been performed appropriately and rigorously? 

Reviewer #1: Yes

3. Have the authors made all data underlying the findings in their manuscript fully available?

Reviewer #1: Yes

4. Is the manuscript presented in an intelligible fashion and written in standard English?

Reviewer #1: Yes

5. Review Comments to the Author

Reviewer #1: The paper is well-written and the authors clearly portrayed their findings.

Additional comments for author

1. The hypotheses developed should be mentioned directly after the text in literature review section, not in the introduction part.

2. In descriptive statistics standard deviation is so high, justify it.

3. Conclusion should highlight the practical implication and theoretical contribution.

6. PLOS authors have the option to publish the peer review history of their article (what does this mean?). If published, this will include your full peer review and any attached files.

Reviewer #1: No

---

## [Author Response · Author response to Decision Letter 0]

2 Aug 2023

Reviewer #1: 

1.The hypotheses developed should be mentioned directly after the text in literature review section, not in the introduction part.

Re: Regarding your comment about the placement of the hypotheses, we have carefully considered your suggestion. Upon reviewing our manuscript, we agree that it would enhance the clarity and organization of our paper to directly mention the hypotheses after the literature review section, rather than in the introduction part. We have made the necessary revisions in the revised manuscript to address this point. Specifically, we have relocated the hypotheses immediately following the literature review section to ensure a more logical flow of information.

By making this adjustment, we believe that the revised manuscript now presents a more coherent structure and better aligns with the standards of academic writing in our field. We appreciate your insightful feedback, which has contributed to improving the overall quality of our manuscript. 

2.In descriptive statistics standard deviation is so high, justify it.

Re: In the modified manuscript, we have added a new paragraph in the revised manuscript to explain why we observed a high standard deviation in the descriptive statistics. We discuss the data sources and collection methods, and provide comparisons and explanations of relevant statistical indicators. Through these additional pieces of information, we aim to enhance the readers’ understanding of the rationality behind the data dispersion.

We believe that this explanation will help clarify and justify our data results, thereby improving the quality and comprehensibility of the paper. Thank you. 

3.Conclusion should highlight the practical implication and theoretical contribution.

Re: In the revised manuscript, we have made revisions to the conclusion section to highlight the practical significance and theoretical contributions. We have reorganized the paragraphs and explicitly mentioned the practical applications of our research and its contributions to theory. Specifically, we have emphasized the impact of our findings on decision-making in developing countries and the role of water management in economic development, as well as its implications for the advancement of theory in related fields.

---

## [Decision Letter · Decision Letter 1]

6 Oct 2023

PONE-D-23-14222R1Does the South-to-North Water Transfer Project Promote the Growth of Enterprises above Designated Size in the Water-receiving Areas? - Evidence from 31 provincial-level administrative regions in ChinaPLOS ONE

Dear Dr. Wang,

Thank you for submitting your manuscript to PLOS ONE. After careful consideration, we feel that it has merit but does not fully meet PLOS ONE’s publication criteria as it currently stands. Therefore, we invite you to submit a revised version of the manuscript that addresses the points raised during the review process.

We look forward to receiving your revised manuscript.

Kind regards,

Chaohai Shen

Academic Editor

PLOS ONE

Additional Editor Comments:

Dear Authors,

The original reviewer is unavailable. Fortunately, we have successfully invited two outstanding reviewers who provided valuable comments on your revision. Please try to make revisions accordingly.

Sincerely,

Reviewers' comments:

Reviewer's Responses to Questions

**Comments to the Author**

1. If the authors have adequately addressed your comments raised in a previous round of review and you feel that this manuscript is now acceptable for publication, you may indicate that here to bypass the “Comments to the Author” section, enter your conflict of interest statement in the “Confidential to Editor” section, and submit your "Accept" recommendation.

Reviewer #2: (No Response)

Reviewer #3: All comments have been addressed

2. Is the manuscript technically sound, and do the data support the conclusions?

Reviewer #2: Yes

Reviewer #3: Partly

3. Has the statistical analysis been performed appropriately and rigorously? 

Reviewer #2: Yes

Reviewer #3: No

4. Have the authors made all data underlying the findings in their manuscript fully available?

Reviewer #2: Yes

Reviewer #3: No

5. Is the manuscript presented in an intelligible fashion and written in standard English?

Reviewer #2: Yes

Reviewer #3: Yes

6. Review Comments to the Author

Reviewer #2: Does the South-North Water Transfer Project have a catalytic effect on the growth of the number of businesses in the receiving area? This is a good idea and an interesting exercise. However, the argumentation process and methodology seems to be not rigorous enough. Here are my main concerns:

（1）Is there a direct link between SNWTP and growth in the number of enterprises? Can this relationship be captured by water supply indicators alone?

（2）Please discuss the specific water users and water consumption of the South-to-North Water Diversion Project, as well as which enterprises use SNWTP water? If the enterprises in the receiving area do not use water from SNWTP, then the statistical analyses in the article are out of touch with reality and the results obtained are wrong.

（3）Why were 31 provinces chosen? Because SNWTP does not cover the whole country.

（4）Why not compare the rate of increase in the number of businesses in the receiving area before and after the water supply?

（5）The manuscript is a macro study, and it seems that similar conclusions can be drawn without the need for such macro statistics.

（6）The charts are not standardised, e.g. there are no names for the horizontal and vertical axes.

（7）The manuscript needs careful editing for typos and language throughout.

Reviewer #3: I believe there is still significant room for improvement in this article. The author appears to have presented the research in a careless manner.

Here are some suggestions for your reference:

1.The introduction section should focus on introducing the research problem of this article and discussing it in the context of previous studies to highlight the value and novelty of this research.

2.The literature review section appears to be lengthy and includes excessive background information. It is important to ensure that the literature review is directly related to the core research questions and provides a comprehensive overview of existing studies in that specific area.

3.Why is there an analysis of the impact of mixed ownership in the section on heterogeneity analysis?

4.The author believes that mixed ownership reform is an important pathway. However, this conclusion is not analyzed in detail within the text, and there is a lack of extensive theoretical analysis to substantiate its validity.

5.The references cited in this article are quite outdated and require updating and supplementing with relevant research from the past three years.

7. PLOS authors have the option to publish the peer review history of their article (what does this mean?). If published, this will include your full peer review and any attached files.

Reviewer #2: No

Reviewer #3: No

---

## [Author Response · Author response to Decision Letter 1]

9 Nov 2023

Dear Editor and Reviewers,

Thank you for your valuable feedback and constructive comments on our manuscript titled “Does the South-to-North Water Transfer Project Promote the Growth of Enterprises above Designated Size in the Water-receiving Areas?” (Manuscript ID: PONE-D-23-14222R1). Following the reviewers' comments, we have modified and improved our manuscript according to your kind advices and referee's detailed suggestions.

Thank you very much for all your help and looking forward to hearing from you soon.

Best regards

Sincerely,

Prof. Chi

Communication University of China

Tingwang@cuc.edu.cn

Reviewers’ comments and our responses are as follows:

Reviewer #2: 

（1）Is there a direct link between SNWTP and growth in the number of enterprises? Can this relationship be captured by water supply indicators alone?

Reply: Thank you very much for your review comments. In the production of large-scale industrial enterprises in water-deficient areas, the importance of water resources has been mentioned in several papers. I have incorporated these references into the first paragraph of the introduction as background information. The South-to-North Water Diversion Project is a direct water transfer project that can quickly change the water supply in the receiving areas. It provides a rapid and direct solution to alleviate the sustainability challenges caused by water resource scarcity. Additionally, through our empirical analysis, we have found a direct and significant correlation between the project's water diversion and the growth of large-scale industrial enterprises. The literatures, combined with our empirical research, supports the existence of this direct relationship. 

Regarding the issue of water supply indicators, the South-to-North Water Diversion Project is a direct water transfer project, and the changes brought about by the water diversion are intuitively reflected in the changes in water supply in the northern region after the project's water diversion. Therefore, the study of this event can be quantitatively analyzed using water supply quantity as an indicator. We have added this point in the second paragraph of the introduction.

In repetition, I would like to express my gratitude for your precious suggestions. Your feedback will help us improve our research further.

（2）Please discuss the specific water users and water consumption of the South-to-North Water Diversion Project, as well as which enterprises use SNWTP water? If the enterprises in the receiving area do not use water from SNWTP, then the statistical analyses in the article are out of touch with reality and the results obtained are wrong.

Thank you for your question. The water from the South-to-North Water Diversion Project, after being mixed with locally-sourced water in a certain proportion at the water plants in the receiving area, will flow from the water stations into every open tap in the receiving area. This information is publicly available on the official website of China's Ministry of Water Resources. In other words, the water from the South-to-North Water Diversion Project will be used by all enterprises, institutions, and households in the receiving area, confirming the scale of this project. We have included this point in the second paragraph of the introduction, and once more, thank you for your priceless question.

（3）Why were 31 provinces chosen? Because SNWTP does not cover the whole country.

Thank you for reviewing our manuscript. Due to the differences in the level and stage of economic development between the northern and southern regions, a direct analysis of growth values and proportions would undoubtedly lead to serious endogeneity issues. Therefore, we have applied a differencing technique to the data from the receiving area and data from other regions nationwide. By removing the individual effects of the receiving area, we obtain a comparison that reflects the net policy effects. We have discussed this point in the section on "Water Supply and the Number of Large-scale Industrial Enterprises." Recurrently, thank you for your appreciable feedback.

（4）Why not compare the rate of increase in the number of businesses in the receiving area before and after the water supply?

Thank you for your suggestions. Your insights are very appreciated. The reason we did not compare the growth rates of the number of enterprises in the receiving area before and after water supply is that the difference-in-differences model has significant advantages when studying the impact of policies with a clearly defined implementation time. It can control for individual fixed effects and time fixed effects, providing more reliable causal inferences than direct comparisons of changes in the treatment group before and after implementation.

Therefore, to investigate this issue from the perspective of applied economics, this study anchored on company-level data for large-scale industrial enterprises and used a difference-in-differences model to observe the practical contributions of the South-to-North Water Diversion Project and explore the mechanisms behind these contributions. 

We have added this discussion about method selection in the last paragraph of the introduction. Thank you for your prized feedback.

（5）The manuscript is a macro study, and it seems that similar conclusions can be drawn without the need for such macro statistics.

Thank you for your review comments. Your points are worth considering. We understand your concerns. However, when studying socio-economic issues, various influencing factors can be complex. To evaluate causal relationships more scientifically, it is best to utilize statistical and mathematical methods, combined with theoretical foundations, to minimize the impact of endogeneity issues. Conclusions obtained from such approaches can approximate causal relationships in reality within a certain range, rather than merely providing a simplistic characterization of a few correlated factors or variables. Therefore, macro-level statistical results are necessary and useful in this type of research. We have included a discussion on this issue in the last paragraph of the introduction. Encore, thank you for your beneficial feedback. Your suggestions have been very helpful to our research.

（6）The charts are not standardised, e.g. there are no names for the horizontal and vertical axes.

Thank you for reviewing and providing feedback on our manuscript. We sincerely apologize for the lack of standardization in the labels of the charts; it was indeed an oversight on our part. We have revised the charts by adding labels to both the horizontal and vertical axes. Simultaneously, modifications have been made to the newly submitted Stata code to ensure that readers can better comprehend the data presented in the charts. These changes aim to enhance the clarity and understanding of our visual representations.

（7）The manuscript needs careful editing for typos and language throughout.

Thank you for pointing out the issues. We have conducted another round of proofreading and have taken measures to ensure the improvement of the language quality of the manuscript to meet the requirements of the journal. Thank you for reviewing our manuscript once again.

Reviewer #3: Here are some suggestions for your reference:

1.The introduction section should focus on introducing the research problem of this article and discussing it in the context of previous studies to highlight the value and novelty of this research.

Thank you for your suggestions on our manuscript. The suggestions you provided are highly reasonable. During the revision process, we have incorporated the suggestions and combined them with previous research to address the challenges faced by enterprises in the water-receiving areas of the South-to-North Water Diversion Project. We have emphasized the significance of the water resource changes brought about by the project in the first paragraph of the introduction. In the second and third paragraphs, we have highlighted the direct impact of the South-to-North Water Diversion on enterprise growth.

Furthermore, we have included discussions on the socioeconomic promotion effects and various cutting-edge academic perspectives in subsequent paragraphs of the introduction. These discussions serve as a starting point to illustrate the value and innovation of exploring the promoting effect of the South-to-North Water Diversion on the growth of large-scale industrial enterprises. This approach aims to help readers better understand our research question and convey the uniqueness and importance of our study.

Reiterating, we sincerely appreciate your beneficial feedback. Your suggestions will greatly help us improve our manuscript.

2.The literature review section appears to be lengthy and includes excessive background information. It is important to ensure that the literature review is directly related to the core research questions and provides a comprehensive overview of existing studies in that specific area.

Thank you for your review and highly regarded feedback. We greatly appreciate your comments on the literature review section. During the revision process, we have made revisions to streamline the literature review section. We have avoided excessive descriptions of the various routes of the South-to-North Water Diversion Project other than the central route. Additionally, we have excluded literature summaries of topics not directly related to the economic benefits of the South-to-North Water Diversion Project, ensuring that the literature review section is directly relevant to the core research question.

Instead, we have provided an overview of existing research on the balance and equilibrium effects of large-scale water infrastructure projects to make it more closely aligned with the main theme and help readers understand the topic.

In addition, we sincerely appreciate your guidance and feedback. Your suggestions are invaluable in helping us improve our manuscript.

3.Why is there an analysis of the impact of mixed ownership in the section on heterogeneity analysis?

Thank you for your question. Compared to other countries in the world, China, especially the northern region, has a higher proportion of state-owned enterprises in the total number of enterprises. In order to address the issue of excessive crowding out of private enterprises by state-owned enterprises in the market, China has been actively implementing ownership reform in recent years. This is a significant development for economic research at the enterprise level. However, there is currently no scholarly analysis of the impact of large-scale water projects from this perspective. Furthermore, we have utilized data at the enterprise level, which is why we treat ownership as a grouping variable in heterogeneity analysis. We have added this point in the final paragraph of the introduction. 

4.The author believes that mixed ownership reform is an important pathway. However, this conclusion is not analyzed in detail within the text, and there is a lack of extensive theoretical analysis to substantiate its validity.

Thank you for your key points of concern regarding our paper. The issues you raised are indeed worth further discussion. In the revised version, we have delved into the importance of mixed-ownership reform in greater detail. We have expanded the relevant content in the literature review section on "The Equilibrium and Equilibrating Effects of Large-Scale Water Infrastructure" and provided broader theoretical analysis in the section on "Possible Influence of SNWDP on the Reform of Enterprise Ownership" in the heterogeneity analysis. We sincerely appreciate your suggestions, as these improvements will enhance the quality and credibility of our paper.

5.The references cited in this article are quite outdated and require updating and supplementing with relevant research from the past three years.

Thank you for carefully reviewing our manuscript and providing significant suggestions. Your points are very pertinent, and in the revision, we have updated and supplemented the relevant research within the past three years to ensure the currency and comprehensiveness of the literature. Once again, we sincerely appreciate your precious feedback, as your suggestions will help improve the quality of our paper.

We have carefully addressed all the comments and suggestions provided by the reviewer. The revised manuscript reflects these changes, and we believe that the modifications have significantly strengthened the overall quality and contribution of the study.

Once again, we express our gratitude to the editor and the reviewers for their time, expertise, and valuable input. We hope that our revised manuscript now meets the requirements for publication in your esteemed journal.

---

## [Decision Letter · Decision Letter 2]

9 Jan 2024

Does the South-to-North Water Transfer Project Promote the Growth of Enterprises above Designated Size in the Water-receiving Areas? - Evidence from 31 provincial-level administrative regions in China

PONE-D-23-14222R2

Dear Dr. Chi,

We’re pleased to inform you that your manuscript has been judged scientifically suitable for publication and will be formally accepted for publication once it meets all outstanding technical requirements.

Kind regards,

Chaohai Shen

Academic Editor

PLOS ONE

Additional Editor Comments (optional):

Dear Authors,

I think you have made revisions based on the reviewers' comments well. Therefore, I would like to accept your paper for publication.

Sincerely,

Reviewers' comments:

Reviewer's Responses to Questions

**Comments to the Author**

1. If the authors have adequately addressed your comments raised in a previous round of review and you feel that this manuscript is now acceptable for publication, you may indicate that here to bypass the “Comments to the Author” section, enter your conflict of interest statement in the “Confidential to Editor” section, and submit your "Accept" recommendation.

Reviewer #2: (No Response)

Reviewer #3: All comments have been addressed

2. Is the manuscript technically sound, and do the data support the conclusions?

Reviewer #2: Partly

Reviewer #3: Yes

3. Has the statistical analysis been performed appropriately and rigorously? 

Reviewer #2: Yes

Reviewer #3: Yes

4. Have the authors made all data underlying the findings in their manuscript fully available?

Reviewer #2: No

Reviewer #3: Yes

5. Is the manuscript presented in an intelligible fashion and written in standard English?

Reviewer #2: No

Reviewer #3: Yes

6. Review Comments to the Author

Reviewer #2: The author's perfunctory answer is obviously not to my satisfaction, so I recommend rejecting the manuscript.

Reviewer #3: In the revised manuscript, the author has responded to the questions I raised before, and the quality of the paper has been improved. It is recommended to accept it.

7. PLOS authors have the option to publish the peer review history of their article (what does this mean?). If published, this will include your full peer review and any attached files.

Reviewer #2: No

Reviewer #3: No

---

## [Editor Report · Acceptance letter]

13 Feb 2024

PONE-D-23-14222R2 

PLOS ONE

Dear Dr. Chi, 

I'm pleased to inform you that your manuscript has been deemed suitable for publication in PLOS ONE. Congratulations! Your manuscript is now being handed over to our production team.

Kind regards, 

on behalf of

Dr. Chaohai Shen 

Academic Editor

PLOS ONE